# The Inflammatory Bridge Between Type 2 Diabetes and Neurodegeneration: A Molecular Perspective

**DOI:** 10.3390/ijms26157566

**Published:** 2025-08-05

**Authors:** Housem Kacem, Michele d’Angelo, Elvira Qosja, Skender Topi, Vanessa Castelli, Annamaria Cimini

**Affiliations:** 1Department of Life, Health and Environmental Sciences, University of L’Aquila, 67100 L’Aquila, Italy; houseneddine.kacembenhajmbarek@student.univaq.it (H.K.); michele.dangelo@univaq.it (M.d.); vanessa.castelli@univaq.it (V.C.); 2Sbarro Institute for Cancer Research and Molecular Medicine, Temple University, Philadelphia, PA 19122, USA; 3Department of Clinical Disciplines, University Alexander Xhuvani of Elbasan, 3001 Elbasan, Albania; elvira.qosja@uniel.edu.al (E.Q.); skender.topi@uniel.edu.al (S.T.)

**Keywords:** neuroinflammaging, insulin resistance, type 2 diabetes mellitus, neuroinflammation, aging

## Abstract

Chronic low-grade inflammation is a hallmark of both metabolic and neurodegenerative diseases. In recent years, several studies have highlighted the pivotal role of systemic metabolic dysfunction, particularly insulin resistance, in shaping neuroinflammatory processes and contributing to impaired cognitive performance. Among metabolic disorders, type 2 diabetes mellitus has emerged as a major risk factor for the development of age-related neurodegenerative conditions, suggesting a complex and bidirectional crosstalk between peripheral metabolic imbalance and central nervous system function. This review aims to explore the cellular and molecular mechanisms underlying the interaction between metabolic dysregulation and brain inflammation. By integrating current findings from endocrinology, immunology, and neuroscience, this work provides a comprehensive overview of how chronic metabolic inflammation may contribute to the onset and progression of neurodegenerative conditions. This interdisciplinary approach could offer novel insights into potential therapeutic strategies targeting both metabolic and neuroinflammatory pathways.

## 1. Introduction

Type 2 diabetes mellitus (T2DM), neuroinflammation, and neurodegenerative disorders such as Alzheimer’s disease (AD) and Parkinson’s disease (PD) are being increasingly identified as a part of a multifactorial pathological network where chronic central and systemic inflammation is the unifying factor [1,2]. T2DM is no longer thought of as a purely metabolic disorder of insulin resistance, hyperglycemia, and pancreatic β-cell dysfunction; indeed, T2DM is a condition of chronic, low-grade systemic inflammation, which is mainly due to visceral adiposity and ectopic lipid deposition that lead to the ongoing release of pro-inflammatory cytokines like Tumor Necrosis Alpha (TNF-α), interleukin 1 beta (IL-1β), and interleukin 6 (IL-6) as well as chemokines, acute-phase proteins, and adipokines like resistin and leptin [3,4,5]. These inflammatory mediators interfere with insulin signaling in the liver, muscles, and adipose tissue, inducing endothelial dysfunction, oxidative stress, and mitochondrial abnormalities, all of which contribute to enhanced metabolic dysregulation [6]. Conversely, neurodegenerative diseases, particularly AD and PD, are characterized by chronic neuronal loss, synaptic dysfunction, and deposition of misfolded proteins such as amyloid-β and α-synuclein [7]. Increasing evidence suggests that these neuropathological features result not only from intrinsic neuronal degeneration but also from chronic neuroinflammation [8,9]. Microglia, the central nervous system (CNS)’s resident immune cells, upon persistent activation by endogenous or systemic inflammatory stimuli, change the phenotype toward a pro-inflammatory M1-like state, which secretes a cascade of neurotoxic cytokines, reactive oxygen species (ROS), and nitric oxide (NO), propagating neuronal injury [10,11,12]. Astrocytes possess the ability under normal conditions to maintain neuronal metabolism by maintaining the integrity of the blood–brain barrier (BBB), but they can switch into a reactive phenotype that disrupts homeostasis and increases inflammatory signaling under adverse conditions [13]. Notably, T2DM has consistently been identified as an independent risk factor for cognitive impairment and dementia, with hyperinsulinemia, advanced glycation end products (AGEs), and inflammatory mediators from the periphery passing through a disrupted blood–brain barrier and triggering central immune responses [14]. In addition, insulin has a direct role in brain function by regulating synaptic plasticity, memory consolidation, and neuronal survival; thus, insulin resistance in the CNS, often described by some authors as “type 3 diabetes,” has been implicated in AD pathogenesis [15]. Clinically, T2DM patients typically have mild neurocognitive dysfunction long before overt neurodegeneration becomes identifiable, suggesting that metabolic derangement may be the forerunner to and predisposition for neurodegenerative cascades. This shared clinical and molecular signature leads one to consider an integrative pathophysiologic model that views metabolic and neurodegenerative diseases, far from being distinct entities, as concomitant manifestations of chronic, inflammatory systemic derangement [14,16]. Examining the inflammatory bridge between peripheral metabolic derangement and central neurodegenerative processes can potentially open up new therapeutic pathways by modulating immune–metabolic crosstalk in order to prevent or delay cognitive decline in diabetic patients [17]. The primary aim of this review is to provide a comprehensive description of the cellular and molecular mechanisms linking IR, systemic inflammation, and neurodegeneration, with particular emphasis on the recently emerging concept of neuroinflammaging (Figure 1).

Through the integration of data from metabolic, immunological, and neuroscientific studies, this article aims to elucidate how metabolic disorders and CNS inflammation are not independent conditions on their own, but rather, interdependent pathophysiologic processes. This perspective is increasingly relevant given the rising global incidence of T2DM and the parallel increase in age-related neurodegenerative diseases such as AD and PD.

Through the analysis of overlapping pathogenic mechanisms, including cytokine signaling, glial activation, disruption of insulin and glucose homeostasis, mitochondrial damage, and oxidative stress, we aim to introduce chronic inflammation as an integrating dimension that interrelates peripheral metabolic disturbances to central neurodegenerative cascades. In addition, the narrative highlights the translational significance of targeting these overlapping pathways in the generation of therapeutic strategies with the potential to counteract both the metabolic and neurological disease components. Particular focus is given to neuroinflammaging as a potential conceptual bridge that may ultimately inform new interventions aimed at modifying disease course in aging populations. Through the elucidation of these mechanistic associations, we propose an integrated model that allows for a transition toward multi-system models of prevention and treatment in age-related disorders.

## 2. Pathophysiology of T2DM: Inflammatory Drivers and Metabolic Consequences

T2DM is a chronic metabolic disorder that afflicts over 500 million individuals across the world, a number set to surge exponentially in the forthcoming two decades as a result of aging populations, sedentary lifestyles, and dietary surplus [18]. It is characterized by sustained hyperglycemia resulting from a combination of insulin resistance (IR) and progressive β-cell dysfunction. IR often precedes the clinical onset of T2DM by several years and represents a pivotal pathophysiological event in the disease continuum [19,20]. At the molecular level, IR is largely fueled by a third-party interaction between genetic susceptibility and environmental stimuli such as obesity, lipotoxicity, and persistent excess nutrients. A characteristic process of IR is the dysregulation of insulin signal transduction pathways, especially the PI3K/Akt pathway, which orchestrates glucose entry and metabolism [21]. Pro-inflammatory signaling molecules such as TNF-α, IL-6, and C-reactive protein (CRP) have been reported to inhibit IRS-1 phosphorylation and activation of Akt, leading to inhibited translocation of GLUT4 to the plasma membrane and decreased glucose uptake [22]. This pro-inflammatory setting is primarily upheld by macrophages and adipose tissue-derived dysfunctional adipocytes secreting a range of cytokines and adipokines, not only suppressing insulin signaling but also causing systemic inflammation [23,24]. Moreover, elevated circulating free fatty acid concentrations cause endoplasmic reticulum stress, oxidative stress, and stress kinase activation of JNK and IKKβ, also suppressing insulin action [25]. Clinically, T2DM is associated with an overabundance of comorbidities, including cardiovascular disease, nephropathy, retinopathy, and neuropathy, most of which are exacerbated by chronic inflammation [26]. Treatments to usual care focus on improving glycemic control and insulin sensitivity by lifestyle intervention and pharmacologic interventions, including metformin (which augments AMP-activated protein kinase activity), GLP-1 receptor agonists, SGLT2 inhibitors, and insulin therapy [27]. These treatments mainly address hyperglycemia and do not fully correct the inflammatory mechanisms responsible for IR and β-cell loss. A growing body of evidence indicates that chronic low-grade inflammation is not just a byproduct but an integral causal component of metabolic impairment, placing T2DM as an immuno-metabolic disease [28]. This inflammatory component is, further, a mechanistic link to neurodegenerative disease, since blood-borne inflammatory mediators have been shown to cross the blood–brain barrier, activate microglia, and alter neurovascular integrity [29]. Thus, an understanding of the molecular and cellular basis of insulin resistance and diabetes within inflammation is critical for shedding light on the broader gradient of inflammation-mediated chronic disease states, including those of the central nervous system [3].

Neuroinflammation has increasingly been recognized as a key pathophysiological feature in numerous neurodegenerative disorders, including AD, PD, and MS, that together afflict hundreds of millions worldwide and impose a significant burden on aging populations [30]. Although CNS inflammation may initially be beneficial, its activation, either chronic or dysregulated, contributes significantly to neurodegeneration. The mechanism is primarily mediated by resident glial cells, astrocytes, and microglia, which become activated by various insults including traumatic damage, infections, or the deposition of misfolded proteins [31]. Upon activation, the cells secrete a vast array of pro-inflammatory mediators like TNF-α, IL-1β, IL-6, ROS, and NO, which collectively disrupt synaptic integrity, compromise the BBB, and enhance neuronal damage [32]. Cumulative exposure to these inflammatory mediators causes a pathologic feedback loop of persistent glial activation, oxidative stress, and chronic neuronal dysfunction. At the molecular level, several signaling cascades are involved in perpetuating and exacerbating this neuroinflammatory response [33,34]. They include the nuclear factor kappa-light-chain-enhancer of activated B cells (NF-κB) pathway, the mitogen-activated protein kinase (MAPK) cascade, and the NOD-like receptor family pyrin domain-containing 3 (NLRP3) inflammasome, which is central to sensing cellular stress and facilitating the maturation of IL-1β and interleukin-18 (IL-18). Concurrently, toll-like receptors (TLRs), including TLR4, recognize damage-associated molecular patterns (DAMPs) and further enhance the innate immune responses within the CNS. Despite ongoing therapy, current pharmacologic treatment of neurodegenerative diseases remains mainly symptomatic [35,36]. Medications such as acetylcholinesterase inhibitors and N-methyl-D-aspartate (NMDA) receptor antagonists are employed in AD, and dopaminergic drugs are the mainstay in PD treatment. These drugs, however, do not affect the pro-inflammatory mechanisms underlying disease progression. To this end, novel anti-inflammatory approaches, ranging from non-steroidal anti-inflammatory drugs (NSAIDs) to microglial modulators and NLRP3 inhibitors, are being explored, though with variable clinical outcomes thus far [37,38]. It is now becoming clear that neuroinflammation is not a secondary consequence of neuronal injury, but rather a primary driver of disease onset and progression. This underscores the need to understand the mechanisms of control of glial activation and to explore how alterations in systemic metabolism may interact with and enhance neuroinflammatory mechanisms [31,39].

## 3. Neuroinflammaging and Metabolic Dysregulation in Aging Brains

The mechanism of neuroinflammaging is the age-related increase in chronic, low-grade inflammation of the CNS and is more and more being recognized as a key mediator of progressive cognitive decline and increased vulnerability to neurodegenerative disease in the elderly [31,40]. This phenomenon is due to the interaction of immunosenescence, the progressive remodeling and immune function decline with age, and chronic neuroinflammatory activation of glial cells, particularly microglia and astrocytes [41]. Glial cells of the aging brain tend toward a pro-inflammatory type and are primed even under no evident pathology. Upon stimulation, this disorder results in an exaggerated inflammatory response characterized by sustained release of such mediators as IL-1β, IL-6, TNF-α, and C-reactive protein (CRP) [42,43]. At the same time, systemic aging is accompanied by elevated levels of circulating inflammatory molecules, a condition described as “inflammaging”, that amplifies peripheral tissue damage and has adverse effects upon the CNS. One of the mechanistically significant links is progressive impairment of BBB integrity that enhances permeability to peripheral cytokines, ROS, and metabolic byproducts, thereby promoting local neuroinflammation [44,45]. One of the numerous systemic factors driving neuroinflammaging is T2DM, which is particularly pertinent as it is characterized by a chronic metabolic and inflammatory load. T2DM, which is typified by IR and hyperglycemia, exerts important effects on immune regulation and is highly correlated with cognitive impairment and increased risk of dementia [46]. Insulin receptors are widely expressed in the neurons and glial cells, where they modulate synaptic plasticity, neuronal metabolism, and survival. In states of IR, impaired insulin signaling compromises glucose uptake, promotes mitochondrial dysfunction, and elevates oxidative stress, contributing to the activation of key inflammatory pathways including NF-κB, MAPKs, and the NLRP3 inflammasome [47,48,49]. In addition, the accumulation of AGEs and their interaction with the receptor for AGEs (RAGE) additionally augment cytokine secretion and immune activation. TLRs, and especially TLR4, are triggered in elderly and insulin-resistant CNS tissue to enhance the recognition of DAMPs and maintain neuroimmune responses [50,51]. These studies showed how neuroinflammaging does not necessarily have to be viewed as an irreversible passive process of aging, but rather, a dynamic and modifiable process shaped by genetic predisposition, exposure to the environment, and metabolic condition. Physical inactivity, overnutrition, and obesity are lifestyle factors contributing to heightened systemic and central inflammation [30,52]. Concurrently, senescent reductions in anti-inflammatory mediators such as interleukin-10 (IL-10) and transforming growth factor beta (TGF-β) impede the resolution of inflammation and enhance chronic neuroimmune activation. Clinically, it manifests as progressive cognitive impairment, neuropsychiatric symptoms, and increased susceptibility to AD, PD, and vascular dementia. So far, therapy for T2DM and neurodegenerative disorders has been siloed and ignores the shared inflammatory pathways at the point of overlap [53,54,55]. Greater insights into neuroinflammaging as a convergent pathological axis offer a powerful rationale for multidisciplinary approaches targeting neuroinflammation in the setting of metabolic derangement. These may include combined lifestyle and pharmacologic interventions aimed at the reversal of CNS insulin resistance, modulation of inflammatory signaling, and preservation of BBB integrity [30,56]. Lastly, neuroinflammaging provides a unifying hypothesis linking systemic metabolic derangement to age-related neurodegeneration and highlights the imperative for an integrative, systems-level approach to disease prevention and management in older populations [57]. The recognition of neuroinflammaging as a clinically relevant phenomenon has been informed by converging experimental and clinical evidence in support of aging having an inextricable link with a subclinical, chronic inflammatory state within the CNS. Initial evidence emerged from neuropathological examinations of post-mortem brain tissue, which showed evidence of microglial activation along with elevated expression of pro-inflammatory cytokines (IL-1β, IL-6, and TNF-α) even in cognitively healthy elderly individuals [42,58]. These findings were corroborated by preclinical models, which had age-related increases in inflammatory mediators, notably in hippocampal regions with pivotal functions in memory processing and synaptic plasticity. Pertinently, systemic inflammatory conditions, among them T2DM-induced ones, have been revealed to facilitate the exacerbation of neuroinflammatory processes, providing evidence for the suggestion of a bidirectional interaction between peripheral metabolic derangement and central immune activation [59,60,61]. Clinically, neuroinflammaging is marked by an increased susceptibility to neurodegenerative disorders like AD and PD as well as an increased risk of cognitive impairment in individuals with metabolic syndrome or T2DM. Augmenting this association, elevated systemic levels of CRP, IL-6, and TNF-α were confirmed as independent predictors of cognitive decline and dementia in longitudinal population-based cohorts [62,63]. These findings heighten the translational relevance of neuroinflammaging and have shattered the paradigm for the treatment of neurodegenerative diseases, putting inflammation not merely as a secondary phenomenon, but as a therapeutic target in itself [54]. Consequently, clinical practice has begun to grow, with increased emphasis on anti-inflammatory treatments that not only minimize neurodegeneration but also impede age-related pathological mechanisms. Elucidation of the etiology and clinical implications of neuroinflammaging provides a basic framework to dissect the molecular mechanisms through which aging, chronic inflammation, and metabolic dysregulation converge to accelerate neurodegeneration [54,64].

## 4. Microglia and Astrocytes in Age-Related CNS Inflammation

Microglia, the immune effector cells resident within the CNS, are vital for cerebral homeostasis, where they perform necessary functions of immune surveillance, synaptic plasticity, and facilitation of repair response following injury. Under healthy conditions, microglia maintain an active equilibrium between tolerance and surveillance, but this is disrupted during aging and in systemic metabolic injury, including IR and T2DM [65,66].

In these contexts, microglia undergo profound phenotypic and functional reprogramming, acquiring a persistently activated, pro-inflammatory profile, a phenomenon commonly referred to as microglial priming. This state is marked by increased expression of surface activation markers, heightened responsiveness to secondary stimuli, and sustained secretion of pro-inflammatory mediators such as IL-1β, IL-6, and TNF-α [65,67]. Age-related alterations in microglia include dystrophic morphology, reduced ramification, and impaired phagocytic efficacy, all reflecting a compromise in homeostatic control. These are also enhanced by peripheral signals for inflammation, most significantly in the scenario of T2DM, wherein elevated levels of circulating adipokines and cytokines traverse a compromised BBB and directly contribute to the activation of central immune responses [68,69]. In parallel, insulin signaling disruption in microglial cells disrupts metabolic flexibility and induces glycolytic shift, mitochondrial dysfunction, and enhanced production of ROS. This metabolic process remodeling sustains the inflammatory phenotype but is also detrimental to neuroprotective activities, ultimately resulting in neuronal injury, synaptic loss, and dysfunctional neurogenesis [70,71]. Collectively, the progressive transformation of microglia into chronically activated cells within the context of aging and metabolic dysfunction is a critical cellular event in the neuroinflammaging process. Through the integration of peripheral metabolic alteration with central immune activation, primed microglia become pivotal effectors in the pathogenesis of age-related cognitive dysfunction and neurodegenerative disease [72].

### 4.1. Microglia-Mediated Inflammatory Responses

Astrocytes, the most abundant glia of the CNS, play a key role in neuronal integrity as they regulate extracellular levels of neurotransmitters, ensure synaptic function, maintain the integrity of the BBB, and provide metabolic and structural support to surrounding neurons. Astrocytes in pathological conditions such as aging and metabolic diseases, i.e., T2DM, exhibit a phenotypic transformation called reactive astrogliosis [13,73]. This pathophysiology is characterized by cellular hypertrophy, transcriptional reprogramming, and the release of pro-inflammatory mediators, which together contribute to the amplification of neuroinflammatory phenomena [74,75]. Astrocytes maintain bidirectional and dynamic communication with microglia, which form a bidirectional signal axis, governing the CNS immune landscape. The pro-inflammatory cytokines IL-1β and TNF-α secreted from activated microglia initiate astrocyte activation and induce increased chemokine and cytokine production, leading to increased microglial reactivity. This initiates a feedback loop to maintain and amplify the inflammatory state. At the same time, IR and chronic hyperglycemia-exposed astrocytes exhibit disturbed glucose metabolism and mitochondrial dysfunction. This, in turn, is translated into the increased generation of ROS and inflammatory transcriptional activation by NF-κB and signal transducer and activator of transcription 3 (STAT3) [13,76,77]. Such molecular alterations compromise the critical homeostatic roles of astrocytes, including glutamate uptake, buffering of potassium, and metabolic support, and in this way, increase the risk of excitotoxicity and neuronal stress. Furthermore, astrocytes also play a vital role in BBB integrity by controlling the tight junctions of the endothelial cells [78,79]. However, in chronic inflammation and diabetes, this regulatory activity is lost, ensuring greater BBB permeability and easier entry of peripheral immune cells into the CNS. The astrocyte–microglia axis is therefore an important node within the neuroinflammaging pathophysiological network. Its central function in sustaining the inflammatory environment of the diabetic and aging brain qualifies it as a therapeutic target for preventing or reducing neurodegenerative events [80,81].

### 4.2. Astrocytes and Their Role in Neuroinflammaging

We have learned how the microglia–astrocyte axis has a central function in the pathogenesis of neurodegenerative disorders. Dysregulation of these cell populations can thus undermine BBB integrity. The structural and functional integrity of the BBB is crucial for upholding the immune-privileged status of the CNS, protecting neural tissue from peripheral immune activation and systemic inflammatory signals. Yet, age and the development of metabolic dysregulations occurring in T2DM and IR are shown to impair the integrity of the BBB, promoting increased permeability and the ingress of circulating immune cells and inflammatory mediators into the brain parenchyma [82,83].

Mechanistically, hyperglycemia, accumulation of AGEs, and elevated systemic levels of IL-6 and CRP account for endothelial dysfunction and disruption of tight junction structure, thereby compromising the barrier’s selectivity [84]. The ensuing breach allows entry of peripheral immune cells into the CNS, where they interact with resident glial populations and exacerbate local inflammatory responses. Infiltrating monocytes also develop macrophage-like phenotypes and release pro-inflammatory cytokines such as TNF-α and IL-1β, which further stimulate microglia and promote neuronal injury [85,86]. Additionally, CD4^+^ T cells have been identified in increased numbers within the aged and diabetic brain, where they contribute to neurodegenerative processes through the secretion of interferon-gamma (IFN-γ) and other immunomodulatory molecules. Chronic systemic inflammation associated with T2DM also modulates the expression of vascular adhesion molecules on brain endothelial cells: intercellular adhesion molecule 1 (ICAM-1) and vascular cell adhesion molecule 1 (VCAM-1) enhance leukocyte adhesion and transmigration into the CNS [87,88]. In addition to increasing neuroinflammation, peripheral immune cell infiltration adds to oxidative stress, mitochondrial dysfunction, and dysregulation of neuronal homeostasis. As such, BBB compromise and peripheral immune cell entry constitute a fundamental cellular axis bridging systemic metabolic inflammation and central neuropathology in the context of neuroinflammaging [89,90].

Clarification of this interface is of pivotal importance to the development of therapeutic strategies aimed at preserving BBB integrity and inhibiting immune-mediated neurodegeneration in the context of aging and metabolic disease.

### 4.3. Inflammatory Signaling Pathways in Neuroinflammaging

The chronic, low-grade neuroinflammatory state that characterizes neuroinflammaging is the outcome of the coordinated activation of several intracellular signaling cascades, which operate to enhance immune responses and perpetuate tissue dysfunction in the aging and metabolically compromised brain. The best characterized and functionally most important of these are the NF-κB, Janus kinase/signal transducer and activator of transcription (JAK/STAT), and the NLRP3 inflammasome pathways. These pathways work together to sustain the inflammatory environment, linking the systemic metabolic dysregulation occurring in IR and T2DM to CNS dysfunction [91,92,93]. The NF-κB pathway is a master regulator of inflammatory gene expression and is highly attuned to metabolic stress. In IR and T2DM, both hyperglycemia and elevated circulating free fatty acids activate toll-like receptors, with downstream signaling cascades leading to the phosphorylation and degradation of the IκB inhibitors and nuclear translocation of NF-κB. Notably, NF-κB signaling intersects with insulin signaling at multiple levels: it negatively regulates insulin receptor substrate (IRS) activity, thereby fostering insulin resistance, while impaired PI3K/AKT signaling (common in insulin-resistant states) cannot suppress NF-κB activity, creating a vicious cycle of inflammation and metabolic disruption [94,95]. Once in the nucleus, NF-κB stimulates the transcription of a wide range of pro-inflammatory mediators, including TNF-α, IL-6, and IL-1β. Aside from its direct transcriptional activity, NF-κB also primes the NLRP3 inflammasome by upregulating the expression of its critical components, generating a feed-forward loop that amplifies and perpetuates neuroinflammatory signaling [96,97,98]. The JAK/STAT pathway is also involved in such an inflammatory milieu via the signal transduction of cytokine- and growth factor-induced signals. In neuroinflammaging, excessive IL-6 and IFN-γ activate JAK kinases, leading to the phosphorylation and dimerization of STAT proteins, and they translocate to the nucleus to regulate pro-inflammatory and immune gene transcription [99,100].

Of interest is the STAT protein STAT3, whose activation is involved in inducing β-site amyloid precursor protein cleaving enzyme 1 (BACE1) upregulation, bridging inflammation and amyloidogenic processes in AD. Conversely, anti-inflammatory cytokines such as interleukin-4 (IL-4) can activate the signal transducer and activator of transcription 6 (STAT6), promoting a reparative microglial phenotype and suggesting that selective JAK/STAT signaling modulation may be therapeutic [101,102]. The NLRP3 inflammasome is a cytosolic cellular stress and DAMPs sensor and requires a priming and an activation signal for full assembly. Priming is usually NF-κB-mediated, while activation is in response to a variety of stimuli, including mitochondrial dysfunction, ROS, and extracellular ATP. Upon activation, NLRP3 recruits the adaptor protein Apoptosis-Associated Speck-like protein containing CARD (ASC) and pro-caspase-1, which results in the cleavage of pro-IL-1β and pro-IL-18 into their active forms and induces pyroptosis, a pro-inflammatory form of programmed cell death. In diabetic conditions, increased mitochondrial ROS production and increased expression of thioredoxin-interacting protein (TXNIP), a known binding partner of NLRP3, further facilitate inflammasome activation, tightly coupling metabolic dysregulation and innate immune signaling [103,104]. Current research has emphasized the role of metabolic reprogramming in controlling these pathways. For instance, glutaminolysis was shown to enhance mitochondrial ROS production in microglia, thereby predisposing to NLRP3 activation. Pharmacological inhibition of glutaminase or induction of mitophagy inhibits this response, reducing oxidative stress and restoring immune homeostasis [105,106]. Furthermore, transcription factor EB (TFEB), a key regulator of autophagy and lysosomal biogenesis, facilitates the turnover of inflammasome components and has been found to participate in the resolution of neuroinflammation. Activation of TFEB has demonstrated beneficial effects in preclinical models of diabetic encephalopathy, highlighting autophagy as a promising therapeutic axis [106,107,108]. Pharmacological inhibitors such as IKKβ inhibitors and selective NLRP3 antagonists (e.g., MCC950) have been shown to dampen neuroinflammatory responses in preclinical neurodegeneration models. Non-pharmacological strategies such as caloric restriction and exercise have also been shown to modulate NF-κB and inflammasome activity by improving mitochondrial efficiency and lowering systemic inflammation [109,110]. In summary, the NF-κB, JAK/STAT, and NLRP3 pathways comprise an interlinked inflammatory circuitry underlying the neuroimmune dysregulation of aging and metabolic disease. Elucidating the crosstalk between these signaling pathways not only informs our understanding of neuroinflammaging but also warrants the development of targeted therapeutics aimed at interrupting these disease circuits to retard neurodegenerative decline.

## 5. Oxidative Stress, Mitochondrial Dysfunction, and Cellular Senescence

As previously detailed, the convergence of oxidative stress, mitochondrial dysfunction, and cellular senescence constitutes a critical pathogenic axis in the development of neuroinflammaging, particularly in the setting of metabolic disturbances such as T2DM and IR. These interrelated processes not only disrupt neuronal function but also actively promote and sustain chronic neuroinflammation within the aging brain [111,112]. Neurons and glial cells, due to their high metabolic activity and limited redox buffering systems, are especially vulnerable to oxidative damage [113,114]. Under hyperglycemic conditions, increased glucose availability enhances glycolytic flux, often driven by HK2 upregulation and the accumulation of glycolytic intermediates. This metabolic overload, along with HK2 displacement from VDAC, may increase mitochondrial membrane potential, forcing the respiratory chain into a reduced state and thus increasing ROS production. In parallel, hyperglycemia suppresses the expression of PINK1–Parkin, impairing mitophagy and exacerbating mitochondrial dysfunction [115]. ROS, in turn, activate redox-sensitive transcription factors such as NF-κB, which drive the expression of pro-inflammatory cytokines and perpetuate the inflammatory cascade [116,117]. We therefore speculate that mitochondrial dysfunction, as both a cause and a consequence of oxidative stress, is a central mediator of neuroinflammaging. Beyond their functions as cellular energy hubs, mitochondria regulate redox homeostasis and immune signaling. mtDNA damage, reduced efficiency of oxidative phosphorylation, and disrupted mitochondrial dynamics are some of the many mitochondrial alterations commonly found in aged or metabolically stressed CNS tissue [118,119]. These dysfunctions lead to the release of mitochondrial-derived damage-associated molecular patterns (mtDAMPs), such as mtDNA, cardiolipin, and mitochondrial ROS, that activate pattern recognition receptors (PRRs), specifically, the NLRP3 inflammasome. This causes activation of caspase-1 and maturation of IL-1β and IL-18, promoting neuroinflammation and cellular stress [120,121]. Meanwhile, the load of senescent cells in the CNS has a key role in the inflammatory milieu. Cellular senescence is a permanent state of proliferative arrest along with metabolic reprogramming and acquisition of a senescence-associated secretory phenotype (SASP), which is the secretion of pro-inflammatory cytokines, chemokines, matrix metalloproteinases (MMPs), and growth factors [122,123]. Senescent astrocytes, microglia, and neurons propagate tissue dysfunction by altering the local microenvironment and inducing paracrine senescence in neighboring cells. Notably, mitochondrial dysfunction is a prominent senescence inducer through mechanisms such as augmented ROS, NAD^+^ depletion, and activation of the DNA damage response (DDR) pathways p53/p21 and p16^INK4a/Rb. Simultaneous activation of NF-κB and mammalian target of rapamycin (mTOR) signaling further augments SASP expression, linking mitochondrial adversity to chronic inflammation [124,125,126]. We can conclude, therefore, that the relationship between mitochondrial dysfunction and cellular senescence is bidirectional and supported by multiple feedback mechanisms. SASP factors can derail mitochondrial turnover through the repression of mitochondrial biogenesis and inhibition of mitophagy, further exacerbating ROS accumulation and metabolic instability [123,127]. In these contexts, sirtuins (particularly SIRT1 and SIRT3) emerge as critical regulators that antagonize these effects by enhancing mitochondrial function and autophagic flux and suppressing inflammatory gene expression. However, sirtuin activity and expression decline with age and in T2DM, favoring the decline in cellular resilience and immune homeostasis [128,129]. Oxidative stress and mitochondrial dysfunction are central molecular processes in the neuroinflammaging process, not only as initiating stimuli but also as amplifiers of chronic inflammatory responses in the aging and metabolically compromised brain. Mitochondria, in addition to their traditional role in ATP synthesis and energy homeostasis, also represent critical nodes for cellular signaling, including the regulation of innate immunity. Their functional impairment, as a consequence, has direct implications for neuroimmune activation and CNS homeostasis [130,131]. Therapeutic strategies that aim to restore mitochondrial integrity and redox balance are becoming increasingly popular. Mitochondrially targeted antioxidants such as MitoQ and SS-31, mitophagy-inducing agents (e.g., urolithin A), and medications that enhance mitochondrial biogenesis via peroxisome proliferator-activated receptor gamma coactivator 1-alpha (PGC-1α) activation are being actively investigated. Moreover, lifestyle interventions such as exercise and caloric restriction have been proven effective in streamlining mitochondrial efficiency and lessening oxidative load in preclinical investigations [132,133]. In summary, oxidative stress and mitochondrial dysfunction are not passive consequences of aging or metabolic disease, but instead are active drivers of neuroinflammaging. Their complex interaction with inflammatory signaling pathways underscores their central role in the promotion of age-related cognitive decline and neurodegeneration. Therapeutic interventions aimed at the replenishment of mitochondrial homeostasis are currently promising insofar as they reduce the detrimental effects of chronic neuroinflammation in aging and metabolic disease. Collectively, oxidative stress, mitochondrial dysfunction, and cellular senescence represent a very interdependent triumvirate that is the cornerstone of the chronic inflammatory condition of neuroinflammaging. Elucidation of the molecular interdependencies among these processes has therapeutic potential in the attempt to restore cellular homeostasis and dampen inflammation-mediated neurodegeneration with aging and metabolic disease [114,118]. A critical downstream consequence of mitochondrial dysfunction and oxidative stress in aging and metabolic diseases is the impairment of autophagy and mitophagy, which in turn exacerbates neuroinflammation and heightens neuronal vulnerability. Autophagy and mitophagy represent evolutionarily conserved cellular quality control mechanisms that are essential for maintaining neuronal homeostasis, particularly in post-mitotic cells such as neurons, where the accumulation of damaged organelles and protein aggregates can have irreversible consequences. Within the pathophysiological framework of neuroinflammaging, these catabolic processes are increasingly recognized as key modulators of neuroimmune balance and mitochondrial integrity [134,135,136]. Autophagy is tightly regulated by nutrient- and energy-sensing pathways, primarily through the activation of AMP-activated protein kinase (AMPK) and the repression of mTOR, which coordinate the induction of autophagosome formation via the ULK1 complex and autophagy-related (ATG) proteins [137,138]. Under aging and IR conditions, autophagic efficiency is often compromised due to chronic mTOR activation, impaired AMPK signaling, and lysosomal dysfunction. This defect results in intracellular accumulation of defective mitochondria, aggregated proteins, and metabolic byproducts such as lipofuscin, cellular stress markers that induce redox imbalance and amplify inflammatory signaling cascades [139]. Mitophagy, selective removal of damaged mitochondria through the autophagy–lysosome pathway, is especially crucial in limiting neuroinflammation. In response to mitochondrial stress, the PTEN-induced kinase 1 (PINK1)–Parkin pathway orchestrates the recognition and degradation of depolarized mitochondria. PINK1 is accumulated on the outer mitochondrial membrane and recruits the E3 ubiquitin ligase Parkin, which ubiquitinates outer membrane proteins to facilitate autophagosome recognition [140,141]. Deficits in PINK1–Parkin signaling in aging brains and diabetic models impair mitophagic flux, allowing damaged mitochondria to survive and release mtDAMPs, which induce NLRP3 inflammasome activation and promote neuroinflammatory responses [141,142]. Aside from organelle turnover, autophagy also plays a critical role in regulating immune signaling by controlling the senescence-associated secretory phenotype (SASP). Autophagy disrupts the enhanced NF-κB activation and overproduction of pro-inflammatory cytokines such as IL-1β, IL-6, and TNF-α, which are involved in generating and perpetuating chronic inflammation. In T2DM and IR, defective autophagic activity exacerbates these responses by limiting the degradation of inflammatory mediators and perpetuating cellular stress [143,144]. Several other recent studies have also characterized other mitophagy receptors like NIX/BNIP3L, BNIP3, and FUNDC1 that control Parkin-independent mitophagy. Dysregulation of these pathways has been implicated in glial activation, neuronal degeneration, and BBB disruption, and hence, further substantiates their involvement in the pathogenesis of neuroinflammaging [145,146,147]. Notably, autophagy and mitophagy manipulation has also been shown to have therapeutic promise. Rapamycin, metformin, spermidine, and urolithin A, along with NAD^+^ precursors, have all been demonstrated to be effective at restoring autophagic flux and suppressing inflammation in preclinical models of both aging and neurodegeneration. In addition, non-pharmacological interventions such as caloric restriction and regular physical activities have been shown to confer beneficial effects on autophagy-regulating pathways and can preserve cognitive ability and immune capacity in the aged and in diabetics [148,149]. Briefly, impaired autophagy and mitophagy are basic mechanisms linking mitochondrial dysfunction, metabolic stress, and neuroinflammation in brain aging. Therapeutic targeting of these degradative pathways promises a dual strategy to both suppress inflammation and support neuronal viability, thereby breaking the self-perpetuating cycle of neuroinflammaging and its downstream neurodegenerative consequences.

## 6. Dysregulated Insulin Signaling in the Brain

Insulin exerts broad regulatory effects throughout the CNS, ranging from glucose metabolism to synaptic plasticity, neuroprotection, cognition, and regulation of neuroinflammatory responses. Over the past decade, the concept of “central insulin resistance”, or impaired insulin action in the brain, has emerged as a central pathophysiological mechanism of neuroinflammation and neurodegeneration, particularly in the context of aging and T2DM [150,151]. In the state of normal physiology, insulin penetrates the BBB using a receptor-mediated transport system and binds to insulin receptors of neurons, astrocytes, and microglia. Ligand binding promotes autophosphorylation of insulin receptors and the recruitment of insulin receptor substrate (IRS) proteins, mostly IRS-1 and IRS-2, which initiate intracellular signal cascades. The PI3K/AKT pathway is a key downstream signaling cascade governing neurotrophic actions, synaptic function, and anti-inflammatory function by inhibiting GSK3β and NF-κB [152,153]. During IR, whether caused by systemic metabolic disturbances, chronic inflammation, or increased age, insulin passage through the BBB is diminished, and expression and phosphorylation of insulin receptors and IRS proteins within CNS tissue are decreased. This disturbance downregulates PI3K/AKT signaling and, at the same time, causes stress-activated kinases such as c-Jun N-terminal kinase (JNK) and inhibitor of kappa B kinase (IKK) to be activated. These phosphorylate IRS-1 on inhibitory serine residues, further impairing insulin signaling. This negative feedback loop promotes glial activation and enhances the production of pro-inflammatory cytokines such as TNF-α and IL-6, which result in exacerbation of neuronal dysfunction [151,154,155]. Specifically, central dysregulated insulin signaling has been implicated in the etiology of AD. Increased GSK3β activity and reduced insulin-degrading enzyme (IDE) levels enhance tau hyperphosphorylation and amyloid-β deposition, two of the molecular characteristics of AD disease. These observations fit with the emerging hypothesis that AD is, in part, a brain type of diabetes, or “type 3 diabetes” [107,156,157]. The neuroinflammatory environment itself can also enhance insulin resistance in the CNS. Pro-inflammatory cytokines TNF-α and IL-1β activate SOCS protein expression and induce endoplasmic reticulum (ER) stress, both of which impair the IR/IRS axis. At the same time, oxidative stress and mitochondrial dysfunction synergistically reduce insulin signaling and amplify inflammatory reactions [23,158]. Therapeutic interventions aimed at improving CNS insulin sensitivity are being increasingly noted for their ability to modulate both cognitive impairment and neuroinflammation. Intranasal insulin delivery, bypassing peripheral insulin resistance, has shown cognitive benefits in patients with mild cognitive impairment (MCI) and early-stage AD. Other drugs such as glucagon-like peptide-1 (GLP-1) receptor agonists and metformin have been shown to have neuroprotection and anti-inflammatory action via the modulation of insulin-related pathways [159,160]. Overall, central insulin resistance is a key molecular interface between metabolic derangement, chronic inflammation, and neurodegeneration in the aging brain. Affecting insulin signaling at this point thus offers a two-faceted therapeutic approach to decrease neuroinflammation and preserve cognitive integrity in neuroinflammaging.

## 7. Therapeutic Implications, Experimental Models, and Future Perspectives

The intricate interrelationship between T2DM, IR, and neuroinflammation underscores the necessity for integrated therapeutic strategies that simultaneously target metabolic dysfunction and chronic inflammation. Traditional compartmentalized approaches may fall short in addressing the multifactorial nature of neuroinflammaging, thereby reinforcing the need for combinatorial interventions capable of modulating shared pathogenic pathways [1,14]. Some anti-diabetic medications, including metformin and GLP-1 receptor agonists, have generated interest not only because of their metabolic actions but also because of their possible neuroprotective and anti-inflammatory actions. Similarly, peroxisome proliferator-activated receptor gamma (PPARγ) agonists such as pioglitazone have been found to possess the ability to modulate neuroinflammation through the inhibition of microglial activation and cytokine secretion. Furthermore, sodium–glucose cotransporter 2 (SGLT2) inhibitors, originally intended for glycemic control, are currently being explored for their potential to mitigate neuroinflammatory reactions and cognitive dysfunction by systemic and central means [161,162,163]. Targeting specific molecular mediators of neuroinflammation has also been found to be meritorious in preclinical models. Inhibitors of NF-κB, NLRP3 inflammasome proteins, and TNF-α have been researched as treatments for their capacity to suppress glial reactivity and reduce neuronal injury [68,164]. Moreover, therapies for the restoration of mitochondrial function, such as coenzyme Q10 analogues, nicotinamide riboside, and mitophagy stimulators such as urolithin A are assessed for their capacity to reverse the oxidative damage and bioenergetic deficits that link IR with neurodegeneration [165]. Experimental models have been instrumental in elucidating the molecular and cellular underpinnings of neuroinflammaging. In vivo models combining diet-induced metabolic alterations with genetic predispositions to neurodegeneration (such as high-fat diet-fed APP/PS1 or 3xTg-AD mice) offer translational platforms to investigate the metabolic–inflammatory–brain axis. These models replicate key features of neuroinflammaging, including cognitive impairment, glial activation, and insulin signaling deficits. Complementary in vitro systems, including neuron/astrocyte/microglia co-cultures, BBB organoids, and brain cell types derived from induced pluripotent stem cells (iPSCs), facilitate high-throughput mechanistic studies and pharmacological screening in controlled environments [166,167].

In the future, integration of precision medicine within the neuroinflammaging model presents significant potential. Identification of molecular biomarkers reflecting metabolic, inflammatory, and genetic risk profiles may enable the early detection of at-risk individuals and guide targeted personalized intervention strategies. Emerging tools such as single-cell RNA sequencing, spatial transcriptomics, and multi-omics platforms are uncovering cell-type-specific regulatory networks governing neuroimmune dynamics in aging brains. Artificial intelligence (AI) and machine learning (ML) approaches are increasingly being employed to deconstruct these complex datasets, ascertain predictive disease trajectories, and optimize therapeutic regimens [168,169,170]. Of utmost significance, intervention timing will likely dictate therapeutic efficacy. Early intervention with prevention can significantly improve clinical outcomes. In this regard, lifestyle modification in the form of regular exercise, nutritional optimization, cognitive stimulation, and control of sleep should be regarded as initial components of any multidimensional therapeutic regimen. Lastly, a synergistic approach integrating pharmacologic agents, behavioral therapies, and evidence-based technologies may provide an integrated template for the overall management of neuroinflammaging and reduction in its contribution to age-related cognitive decline.

## 8. Discussion and Conclusions

The convergence of IR, T2DM, neuroinflammation, and aging defines a multifaceted pathological axis with substantial clinical implications (Table 1).

As discussed in this review, chronic low-grade inflammation emerges as an important mechanistic link between peripheral metabolic dysregulation and CNS degeneration. From early peripheral insulin resistance and systemic inflammatory priming to microglial activation and gradual neuronal injury, crosstalk between immune and metabolic pathways appears to underline much of age-related cognitive decline and neurodegenerative disease, including AD and AD-related dementias.

We have detailed the key cellular and molecular processes that regulate this complex interplay, identifying events such as glial activation, BBB breakdown, mitochondrial dysfunction, oxidative stress, and the development of the senescence-associated secretory phenotype (SASP). Key inflammatory mediators and signaling pathways such as NF-κB, NLRP3 inflammasome, JNK, disrupted insulin signaling, and cytokines such as IL-6 and TNF-α have been implicated in sustaining the inflammatory environment of the aging brain. The neuroinflammaging theoretical framework provides a valuable window through which these multifactorial events can be interpreted and synthesized within a systems-level context. Despite significant advances, there are several major issues that remain to be resolved. Much of the mechanistic insight is derived from animal models or reduced in vitro systems that may not necessarily recapitulate the complexity of human physiology. It has also proven difficult to move these findings into clinically relevant treatments, partly due to the heterogeneity of human populations, comorbidities, and genetic and exposure variability. New therapies are, however, being created. These include metabolic regulators with anti-inflammatory effects and mitochondrially targeted drugs that aim to restore bioenergetic and redox homeostasis. In parallel, the integration of high-resolution omics modalities, neuroimaging platforms, and artificial intelligence (AI)-enabled analytics has the potential to accelerate biomarker discovery and personalized therapeutic development. Continued refinement of experimental models, particularly those derived from human cells, will also be critical to advance translational efforts. Looking ahead, the discipline will be well served by an interdisciplinary approach that brings together endocrinology and neuroscience, clarifying the temporal kinetics of neuroinflammaging, defining early and strong biomarkers of risk, and tailoring interventions to individuals’ inflammatory and metabolic signature will be the keys to lessening the neurodegenerative burden in the aging population. Ultimately, targeting neuroinflammaging promises not only to delay cognitive dysfunction but also to enhance overall health span and quality of life in the elderly.

## Figures and Tables

**Figure 1 ijms-26-07566-f001:**
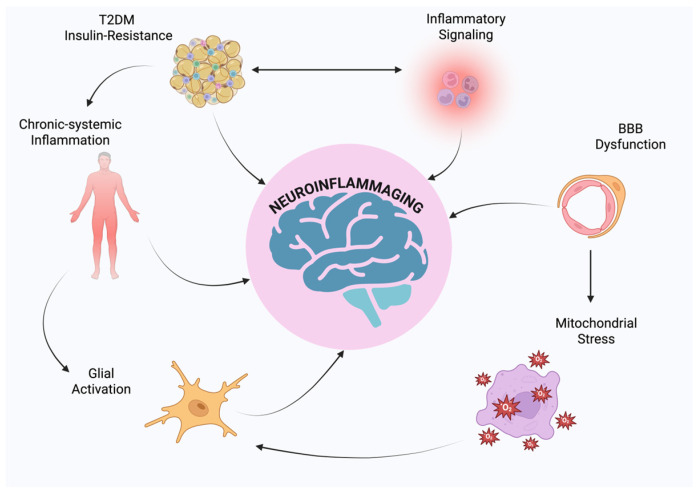
Schematic representation of neuroinflammaging.

**Table 1 ijms-26-07566-t001:** Key pathophysiological axes driving neuroinflammaging, highlighting major molecular players, effects, and relevance to age-related neuroinflammation.

Pathophysiological Axis	Key Players/Pathways	Effects	Relevance to Neuroinflammaging
Metabolic dysfunction	Insulin resistance (IR) T2DM Hyperglycemia	Impaired insulin signaling Chronic low-grade inflammation Lipotoxicity	Triggers systemic inflammation; primes CNS for neurodegeneration
Systemic inflammation	TNF-α, IL-6, IL-1β, CRP, AGEs Adipokines (resistin, leptin)	Endothelial dysfunction BBB permeability Cytokine overflow into CNS	Drives neuroimmune activation and glial priming
Glial activation	Microglia (M1 phenotype) Astrocytes (reactive gliosis)	ROS, NO, cytokine release Impaired neuronal support	Amplifies neuroinflammation and propagates neuronal injury
Mitochondrial dysfunction and oxidative stress	mtROS mtDNA damage SIRT1/3 downregulation	Energy failure NLRP3 activation Senescence	Self-reinforcing loop of neuroinflammation and cell stress
Inflammatory signaling	NF-κB JAK/STAT NLRP3 inflammasome	Pro-inflammatory gene transcription Cytokine maturation Pyroptosis	Central molecular axis connecting metabolic stress and neurodegeneration
Blood–brain barrier dysfunction	Tight junction loss ICAM-1/VCAM-1 upregulation	Immune cell infiltration Loss of CNS immune privilege	Permits systemic factors to influence CNS inflammation
Central insulin resistance	Impaired IR/IRS-1/AKT signaling GSK3β activation Reduced IDE	Β-amyloid accumulation Tau hyperphosphorylation Glial reactivity	“Type 3 diabetes” model of Alzheimer’s pathogenesis

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
