# Peer review of "The Inflammatory Bridge Between Type 2 Diabetes and Neurodegeneration: A Molecular Perspective"

_ijms, 2025, doi:10.3390/ijms26157566_

Round 1
Reviewer 1 Report
Comments and Suggestions for Authors
In this review the authors detailed key cellular and molecular processes regulating aging brain inflammatory environment, associated to T2DM and IR, including glial activation, BBB breakdown, mitochondrial dysfunction, oxidative stress, and senescence-associated secretory phenotype. The authors also emphasized that comprehensive therapy regimens targeting metabolic diseases may center on the intricate link among T2DM, IR, and neuroinflammation.
The manuscript is interesting and both neurobiologists and physicians may find the manuscript to be intriguing and helpful.
I selected a few points as follows:
Line 397 and 398:
The Authors stated that: “In hyperglycemic conditions, excess glucose metabolism leads to electron transport chain overload, enhancing mitochondrial ROS production”. This sentence appears to be too speculative.
Indeed, hyperglycemia inhibits mitochondrial function by suppressing Parkin/PINK1 expression. In addition, hyperglycemia upregulates Hexokinase-2 producing unscheduled increased glucose metabolism. This produces abnormal increases in glycolytic intermediates or glycolytic overload, driving cell dysfunction and vulnerability.
However, HK2 displacement from VDAC can also increases mitochondrial membrane potential, forcing the respiratory chain complexes into a reduced state and increasing electron leakage and ROS formation.
Probably the Authors can ameliorate this aspect.
I would like to suggest checking and rephrasing some sentences for better readability. For example in line 498:
"In addition, caloric restriction and exercise, non-pharmacological interventions, exert beneficial…"
Would be better:
"In addition, non-pharmacological interventions such as caloric restriction and exercise have beneficial effects…"
Other minor points:
line 63:
“Molecularsignature” → “Molecular signature”
line 97:
“underlying inflammatory mechanisms underlying IR and β-cell loss.” → Is it better to avoid repetition?
line 302:
“metabolic dysregulations such as T2DM and IR” → Since the paper is entirely devoted to this condition, it would probably be better to be more direct. For example: metabolic dysregulations occurring in T2DM and IR.
line 334:
The same situation applied to line 302
line 348:
“via the transmission of” → Is it intended to be transduction?
line 394:
The same situation applied to line 302
Author Response
Reviewer 1:
In this review the authors detailed key cellular and molecular processes regulating aging brain inflammatory environment, associated to T2DM and IR, including glial activation, BBB breakdown, mitochondrial dysfunction, oxidative stress, and senescence-associated secretory phenotype. The authors also emphasized that comprehensive therapy regimens targeting metabolic diseases may center on the intricate link among T2DM, IR, and neuroinflammation.
The manuscript is interesting and both neurobiologists and physicians may find the manuscript to be intriguing and helpful.
I selected a few points as follows:
RESPONSE: We sincerely thank the Reviewer for the positive and encouraging evaluation of our manuscript. We are truly grateful for the thoughtful and generous comments highlighting the scientific value and clinical relevance of our review. Knowing that the work may be of interest to both neurobiologists and physicians is particularly rewarding for us, as it reflects our intent to bridge basic science with translational perspectives.
Below, we address the Reviewer’s comments in detail and outline the changes made to the manuscript accordingly.
Line 397 and 398:The Authors stated that: “In hyperglycemic conditions, excess glucose metabolism leads to electron transport chain overload, enhancing mitochondrial ROS production”. This sentence appears to be too speculative.
Indeed, hyperglycemia inhibits mitochondrial function by suppressing Parkin/PINK1 expression. In addition, hyperglycemia upregulates Hexokinase-2 producing unscheduled increased glucose metabolism. This produces abnormal increases in glycolytic intermediates or glycolytic overload, driving cell dysfunction and vulnerability.
However, HK2 displacement from VDAC can also increases mitochondrial membrane potential, forcing the respiratory chain complexes into a reduced state and increasing electron leakage and ROS formation.Probably the Authors can ameliorate this aspect.
Response: We thank the Reviewer for this valuable clarification. We agree that the original sentence was reductive and have now rephrased it to better reflect the multiple converging mechanisms linking hyperglycemia to oxidative stress and mitochondrial dysfunction. The revised sentence integrates HK2-mediated glycolytic overload, increased mitochondrial membrane potential, ROS production, and suppression of PINK1/Parkin-dependent mitophagy.
I would like to suggest checking and rephrasing some sentences for better readability. For example in line 498:"In addition, caloric restriction and exercise, non-pharmacological interventions, exert beneficial…"
Would be better:"In addition, non-pharmacological interventions such as caloric restriction and exercise have beneficial effects…"
RESPONSE: We totally agree with the reviewer and made changes as suggested.
Other minor points:
line 63:“Molecularsignature” → “Molecular signature”
line 97:“underlying inflammatory mechanisms underlying IR and β-cell loss.” → Is it better to avoid repetition?
line 302:“metabolic dysregulations such as T2DM and IR” → Since the paper is entirely devoted to this condition, it would probably be better to be more direct. For example: metabolic dysregulations occurring in T2DM and IR.
line 334:The same situation applied to line 302
line 348:“via the transmission of” → Is it intended to be transduction?
line 394:The same situation applied to line 302
RESPONSE: We sincerely thank the Reviewer for the careful attention to detail and the helpful suggestions to improve readability and precision. We have accepted all the proposed changes and have revised the manuscript accordingly. These include:
- Rephrasing for improved readability (e.g., line 498);
- Correction of a typographical error (“Molecularsignature” in line 63);
- Elimination of redundant wording (line 97);
- Use of more direct phrasing in references to T2DM and IR (lines 302, 334, 394);
- Replacing “transmission” with “signal transduction” for scientific accuracy (line 348).
We believe these edits contribute to a clearer, more accurate, and more polished manuscript, and we are grateful for the Reviewer’s constructive insights.
Reviewer 2 Report
Comments and Suggestions for Authors
The reviewer had some concerns for the structure of the present review article.
Abbreviations should be spelled-out at first appearance and thereafter only abbreviations should be used.
In section 1. Introduction; the authors described the purpose of the present review in 1.1. The aim would be better to be written the last of Introduction; therefore, the order of 1.1. and 1.2. should be reconsidered.
The title of 3.1. was exactly same with 2.1. It might be error and should be corrected.
In section 2, the subsection was only 2.1. and 2.2. was not present. Therefore, subsection did not needed or the sectioning of the content should be reconsidered.
Also in section 4, 4.2. was absent but was 4.1. needed? Moreover, what did "4.1. Role of autophagy and mitophagy in neuroinflammaging" in the section "4. Oxidative stress, mitochondrial dysfunction, and cellular senescence" mean? The relationships should be written in text. And, the sectioning of the content should be reconsidered.
Author Response
Reviewer 2:
The reviewer had some concerns for the structure of the present review article.
Abbreviations should be spelled-out at first appearance and thereafter only abbreviations should be used.
In section 1. Introduction; the authors described the purpose of the present review in 1.1. The aim would be better to be written the last of Introduction; therefore, the order of 1.1. and 1.2. should be reconsidered.
The title of 3.1. was exactly same with 2.1. It might be error and should be corrected.
In section 2, the subsection was only 2.1. and 2.2. was not present. Therefore, subsection did not needed or the sectioning of the content should be reconsidered.
Also in section 4, 4.2. was absent but was 4.1. needed? Moreover, what did "4.1. Role of autophagy and mitophagy in neuroinflammaging" in the section "4. Oxidative stress, mitochondrial dysfunction, and cellular senescence" mean? The relationships should be written in text. And, the sectioning of the content should be reconsidered.”.
RESPONSE:
We sincerely thank the Reviewer for the constructive comments and for carefully reviewing the structure and formatting of our manuscript. We appreciate the suggestions aimed at improving clarity, consistency, and readability. Below, we provide a point-by-point response to each comment and describe the revisions made accordingly.
- Abbreviations: We reviewed the entire text to ensure that all abbreviations are properly spelled out at their first appearance (e.g., type 2 diabetes mellitus (T2DM), reactive oxygen species (ROS), blood-brain barrier (BBB), central nervous system (CNS), etc.), and consistently used in their abbreviated form thereafter.
- Introduction structure (sections 1.1 and 1.2): We have reorganized the introduction so that the aim of the review now appears at the end of the introductory section, as recommended. Subsections 1.1 and 1.2 have been restructured accordingly to improve logical flow.
- Duplicate section title (3.1 and 2.1): We corrected the title of section 3.1, which was mistakenly identical to section 2.1. The new heading now better reflects the content of the subsection.
- Section 2 formatting: Since section 2 contained only one subsection (2.1), we removed the subsection heading and integrated the content directly under the main section title.
- Section 4 formatting and clarity: We revised section 4 by removing the isolated 4.1 heading and integrating the discussion on autophagy and mitophagy into the main body of the section. We also improved the transitions to clarify the relationship between mitochondrial dysfunction, oxidative stress, cellular senescence, and impaired autophagy.
We would like to express our sincere gratitude for your valuable contribution to the review of our article. Your comments and suggestions have undoubtedly enriched our work and guided us in improving the quality and clarity of our scientific contribution. We have carefully considered each of your observations and have made the necessary corrections to ensure the accuracy and completeness of our work.
Round 2
Reviewer 2 Report
Comments and Suggestions for Authors
The reviewer understood that the authors revised the manuscript according to the reviewer's comments. However, the reviewer recommend reconsidering the titles of section and subsection again. The title of section should cover the whole content of the section including the subsections; for example, "3. Microglia" and "3.1. Astrocytes and..." looked separated or completely different. The heading covering the whole contents should be reconsidered as the title of section, and the first paragraph would be divided as the first subsection.
Again, the structure of the present review should be rearranged.
Author Response
We sincerely thank you for your valuable comments and suggestions. We have carefully addressed all the points raised and made the corresponding modifications in the revised version of the manuscript. Thank you again for all the time spent revising our research article and the comments provided that greatly improved our manuscript.